# Optimizing the Empirical Parameters of the Data-Driven Algorithm for SIF Retrieval for SIFIS Onboard TECIS-1 Satellite

**DOI:** 10.3390/s21103482

**Published:** 2021-05-17

**Authors:** Chu Zou, Shanshan Du, Xinjie Liu, Liangyun Liu, Yuyang Wang, Zhen Li

**Affiliations:** 1Key Laboratory of Digital Earth Science, Aerospace Information Research Institute, Chinese Academy of Sciences, Beijing 100094, China; zouchu20@mails.ucas.ac.cn (C.Z.); liuxj@radi.ac.cn (X.L.); liuly@radi.ac.cn (L.L.); 2College of Resources and Environment, University of Chinese Academy of Sciences, Beijing 100049, China; 3Institute of Remote Sensing Satellite, China Academy of Space Technology, Beijing 100095, China; jamiewang0@gmail.com (Y.W.); zhenli.irss@gmail.com (Z.L.)

**Keywords:** solar-induced chlorophyll fluorescence (SIF), data-driven algorithm, Terrestrial Ecosystem Carbon Inventory Satellite (TECIS-1), parameter optimization

## Abstract

Space-based solar-induced chlorophyll fluorescence (SIF) has been widely demonstrated as a great proxy for monitoring terrestrial photosynthesis and has been successfully retrieved from satellite-based hyperspectral observations using a data-driven algorithm. As a semi-empirical algorithm, the data-driven algorithm is strongly affected by the empirical parameters in the model. Here, the influence of the data-driven algorithm’s empirical parameters, including the polynomial order (n_p_), the number of feature vectors (n_SV_), the fluorescence emission spectrum function, and the fitting window used in the retrieval model, were quantitatively investigated based on the simulations of the SIF Imaging Spectrometer (SIFIS) onboard the First Terrestrial Ecosystem Carbon Inventory Satellite (TECIS-1). The results showed that the fitting window, n_p_, and n_SV_ were the three main factors that influenced the accuracy of retrieval. The retrieval accuracy was relatively higher for a wider fitting window; the root mean square error (RMSE) was lower than 0.7 mW m^−2^ sr^−1^ nm^−1^ with fitting windows wider than 735–758 nm and 682–691 nm for the far-red band and the red band, respectively. The RMSE decreased first and then increased with increases in n_p_ range from 1 to 5 and increased in n_SV_ range from 2 to 20. According to the specifications of SIFIS onboard TECIS-1, a fitting window of 735–758 nm, a second-order polynomial, and four feature vectors are the optimal parameters for far-red SIF retrieval, resulting in an RMSE of 0.63 mW m^−2^ sr^−1^ nm^−1^. As for red SIF retrieval, using second-order polynomial and seven feature vectors in the fitting window of 682–697 nm was the optimal choice and resulted in an RMSE of 0.53 mW m^−2^ sr^−1^ nm^−1^. The optimized parameters of the data-driven algorithm can guide the retrieval of satellite-based SIF and are valuable for generating an accurate SIF product of the TECIS-1 satellite after its launch.

## 1. Introduction

Traditional vegetation monitoring processes commonly make use of indicators like vegetation indices [1], leaf area index (LAI) [2], and aboveground biomass [3], etc. Recently, solar-induced chlorophyll fluorescence (SIF) has been regarded as a new way of estimating gross primary productivity (GPP) due to its close correlation with photosynthesis [4]. SIF retrieved from satellites has wide coverage and a large number of long-term measurements, which are expanding its application field [5,6,7,8,9,10].

As a common method used for SIF retrieval, the data-driven algorithm has been widely applied for SIF retrieval from satellite-based hyperspectral data. Guanter et al. [11] first presented a singular vector decomposition (SVD) data-driven algorithm to retrieve SIF from the Greenhouse gases Observing Satellite (GOSAT) space measurements with high spectral resolution (~0.04 nm). Following this, Joiner et al. [12] designed a principal component analysis (PCA) data-driven algorithm to map SIF from the Global Ozone Monitoring Experiment–2 (GOME–2) satellite data with moderate spectral resolution (~0.5 nm). Furthermore, Kohler et al. [13] presented a simplified forward linear model by estimating the atmospheric upward transmittance in advance and used a backward elimination algorithm to automatically determine the parameters of the data-driven algorithm, which was successfully applied for accurately retrieving satellite-based SIF from GOME-2 and the Scanning Imaging Absorption SpectroMeter for Atmospheric Chartography (SCIAMACHY) data. Moreover, a series of global satellite-based SIF products were generated by the data-driven algorithm using spectral measurements from the TROPOspheric Monitoring Instrument (TROPOMI) [14], Orbiting Carbon Observatory 2 (OCO-2) [15], OCO-3 [16], and Chinese Carbon Dioxide Observation Satellite Mission (TanSat) [17]. However, as was mentioned by Kohler et al. [13], there are inconsistencies between different products, even between products generated from the same satellite. Therefore, the accuracy of the satellite-based SIF datasets is dependent on both the specifications of sensors (spectral resolution, spectral range, and the signal-to-noise ratio (SNR), etc.) and the empirical parameters in the data-driven algorithms (fitting window, number of feature vectors, and polynomial order, etc.). Thus, the setting of the algorithm parameters for a specific satellite is very important. To date, simulated satellite data have been extensively used to verify the accuracy and precision of the SIF retrieval algorithm [12,13,18,19,20,21], but it is still not well understood how and how much the empirical parameters in the data-driven algorithm affect satellite-based SIF retrievals.

In this study, we aimed to investigate the influences of the empirical parameters of the data-driven algorithm on satellite-based SIF retrieval and develop an optimized data-driven algorithm for the upcoming First Terrestrial Ecosystem Carbon Inventory Satellite (TECIS-1) with a SIF Imaging Spectrometer (SIFIS) onboard, which will be launched at the beginning of 2022 (details listed in Du et al., 2020 [18]). For this purpose, we generated simulation datasets according to the spectral characteristics of SIFIS (Table 1).

So far, little research has focused on the optimization of empirical parameters for SIF retrieval on TECIS-1, especially for red band SIF retrieval, and a detailed evaluation of empirical parameters is necessary to provide an accurate algorithm and generate satellite-based SIF products after the launch of TECIS-1. For this purpose, the simulation datasets were generated (Section 2.1) to investigate the influence of the empirical parameters of the data-driven algorithm on satellite-based SIF retrieval, including the influence of the fitting window (Section 3.1.1), the number of feature vectors, and polynomial order (Section 3.1.2), as well as the fluorescence spectrum function. The optimal parameter setting for the data-driven SIF retrieval algorithm was determined according to the specifications of the SIFIS onboard TECIS-1, and end-to-end retrievals were conducted to show the precision of the optimal combination of parameters (Section 3.2). The mechanism of the influence of empirical parameters on the retrieval is explained in the Discussion section.

## 2. Materials and Methods

### 2.1. Simulation Datasets

Assuming that the surface is Lambertian, the top-of-atmosphere (TOA) radiance over a vegetation target can be expressed as the following radiative transfer equation [22,23]:(1)$$L_{\mathrm{TOA}}=L_{0}+\frac{L_{\mathrm{TOC}}\cdot \rho_{s}\cdot T_{↑}}{1-S\cdot \rho_{s}}+\frac{\mathrm{SIF}\cdot T_{↑}}{1-S\cdot \rho_{s}}$$
where L_0_ is the atmospheric path radiance, L_TOC_ is the radiance reaching the surface, ρ_s_ is the surface reflectance, S is the atmospheric spherical albedo, T_↑_ is the upward atmospheric transmittance, and SIF is the emitted fluorescence signal at the top-of-canopy (TOC), which should be omitted when simulating radiance over the non-vegetated surface.

The Soil Canopy Observation Photosynthesis and Energy (SCOPE) [24] model was employed to simulate the emitted fluorescence signals and reflectance of vegetation under 126 different canopy structures and leaf biochemic characteristics. Ten different reflectance spectra over snow and soil surface were derived from the spectral library of the Environment for Visualizing Images (ENVI) [25]. Three constant reflectance spectra were set in the Moderate-resolution atmospheric TRANsmission (MODTRAN) [26,27] model to calculate the atmospheric radiation transfer parameters (L_0_, L_TOC_, and S) under 1280 different atmospheric and observation conditions. The input parameters of the two models are shown in Table 2. Figure 1 shows the simulated canopy SIF spectra of different canopy and leaf conditions in the range of 640–800 nm. The fluorescence spectra have a bimodal distribution, with a more obvious peak at around 740 nm.

The spectra of the atmospheric radiation transfer parameters (L_0_, L_TOC_, S, and T_↑_) simulated by MODTRAN have a spectral resolution of 0.005 nm. The canopy SIF and the reflectance spectra over vegetated and non-vegetated surfaces were then resampled to the same interval. Then, the resampled SIF and reflectance spectra were used to determine the TOA radiance spectra according to Equation (1). Finally, 161,280 simulated radiance spectra over the vegetated surface were generated as the test dataset, and 12,800 simulated radiance spectra over the non-vegetated surface were generated as the training dataset.

Simulated radiance spectra with a high spectral resolution of 0.005 nm were then convolved and resampled according to the specifications of SIFIS (a spectral resolution of 0.3 nm and a sampling interval of 0.1 nm). Figure 2 shows a set of TECIS-like simulated spectra, including the canopy radiance spectra (Rad_Veg), solar irradiance spectra (Solar_Irr), upward atmospheric transmittance spectra (T_u), and SIF signal at the top-of-canopy and atmosphere. As can be seen in Figure 2, the absorption lines are obvious under a spectral resolution of 0.3 nm, and the radiance spectra are affected by the absorption of solar atmosphere and Earth’s atmosphere.

Since SIF is a weak signal, its retrieval is very sensitive to noise [13,28,29]. For satellite-based SIF retrieval, the SNR of the instrument is also taken into consideration. The noise added to each radiance signal can be expressed as [30]:(2)$$\mathrm{noise}\left(L_{\mathrm{TOA}},\lambda \right){=\mathrm{noise}}_{r}\cdot \frac{L_{\mathrm{TOA}}\left(\lambda \right)}{\mathrm{SNR}\left(L_{\mathrm{TOA}},\lambda \right)}$$
where λ is the wavelength, noise_r_ is a base noise that obeys a standard normal distribution across the spectra, and SNR is a function of the radiance and wavelength, denoted as [14]:(3)$$\mathrm{SNR}\left(L_{\mathrm{TOA}},\lambda \right){=\mathrm{SNR}}_{\mathrm{ref}}\sqrt{\frac{L_{\mathrm{TOA}}\left(\lambda \right)}{L_{{\mathrm{TOA}}_{\mathrm{ref}}}}}$$
where SNR_ref_ is the reference SNR at the reference radiance level $L_{{\mathrm{TOA}}_{\mathrm{ref}}}$.

Based on the specifications of SIFIS, we set the SNR_ref_ (denoted as SNR below) of 300, 350, 400, 450, and 500 to analyze the influence of noise on empirical parameter optimization. In addition, an SNR of 322 was set to select optimal parameters for SIFIS data.

### 2.2. The Data-Driven SIF Retrieval Algorithm

The first two terms in Equation (1) are non-fluorescent contributions, while the last term is the fluorescent contribution. According to the basic idea of the data-driven algorithm, the non-fluorescence contribution of L_TOA_ is a combination of high-frequency information (including Fraunhofer lines and atmospheric absorption lines) and low-frequency information. The high-frequency part (caused by the absorption of the solar atmosphere and Earth’s atmosphere) has a specific pattern, which can be extracted from the training dataset composed of non-vegetated targets and reconstructed with a small number of feature spectra. The low-frequency part (caused by atmospheric radiation and surface reflection) can be expressed by a polynomial of wavelength (λ) [11,19]. Moreover, SIF spectra usually have a specific shape and can be expressed by a mathematical function. Thus, Equation (1) can be rewritten as:(4)$$L_{\mathrm{TOA}}=\left(\overset{n_{p}}{\underset{i=0}{\sum }}a_{i}\cdot \lambda^{i}\right)\cdot \left(\overset{n_{\mathrm{SV}}}{\underset{j=1}{\sum }}\alpha_{j}\cdot v_{j}\right)+F_{S_{0}}\cdot h_{F}\cdot T_{↑}$$
where $v_{j}$ is the feature vector decomposed by the singular vector decomposition (SVD) method, $a_{i}$ and $\alpha_{j}$ are the coefficients of the polynomial and the singular vectors, respectively, $n_{p}$ is the order of polynomial, $n_{\mathrm{SV}}$ is the number of selected singular vectors, F_s0_ is the SIF signal at the specified wavelength, and T_↑_ is the upward atmospheric transmittance. $h_{F}$ is expressed as a Gaussian function with an average of $\mu_{h}$ and a standard deviation of $\sigma_{h}$:(5)$$h_{F}=\exp\left[\frac{-{\left(\lambda -\mu_{h}\right)}^{2}}{2\sigma_{h}{}^{2}}\right]$$

According to Kohler et al. [13], the upward atmospheric transmittance T_↑_ can be estimated before retrieval, so when n_p_, n_SV_, μ_h_ and σ_h_ were determined, only $a_{i},\;\alpha_{j},\;\mathrm{and}\;F_{S_{0}}$ were left as unknown parameters; then, the retrieval could be turned into a linear least-squares problem.

The parameters mentioned above are mostly set by experience. For example, μ_h_ is generally 740 nm for far-red fluorescence, and 692 nm for red fluorescence, n_p_ is generally 2–4. However, the influence of different parameters on the results remains to be analyzed. In addition to parameters in the forward model, the fitting window, i.e., the spectral range in Equation (5), is also a key parameter, which determines the number of absorption lines used. Meanwhile, because the random noise of the instrument also fills the absorption line, it interferes with the retrieval so different SNR levels were also considered in this study.

For red SIF retrieval, the spectral range covered by O_2_-B absorption lines (682–697 nm) covered nearly all the fitting windows selected in previous studies [18,20,31]. For far-red SIF retrieval, the water vapor absorption band (range from 712 nm to 745 nm), the O_2_-A band (range from 755 nm to 775 nm), and the atmospheric window between them were commonly used. Thus, we completed a series of experiments to preliminarily select the range of the far-red fitting window (Table 3). Table 3 shows the retrieval results with the minimum RMSE in the fitting windows after trial and error. Fitting windows within the O_2_-A band were used in Exp. 1 and 2, Exp. 3–6 used the fitting window containing water vapor absorption lines, while the fitting window of Exp. 5 and 6 also contained the atmospheric window range from 747 nm to 758 nm. The results showed that retrievals using a fitting window containing oxygen absorption lines (755–778 nm and 747–780 nm) or merely water vapor absorption lines (724–747 nm and 715–748 nm) are not as good as using a fitting window that combines water vapor absorption lines and the atmospheric window (735–758 nm and 720–758 nm), which is also true for the fitting windows of the same width (735–758 nm, 724–747 nm and 755–778 nm). Exp. 6–14 showed that with the adjustment of n_p_ and n_SV_, reasonable retrievals (with RMSE less than 1 mW m^−2^ sr^−1^ nm^−1^) could be achieved when n_p_ was between 1–5 and n_SV_ was less than 20. Through the optimization, the RMSE of SIF retrieval can be reduced from 1.18 mW m^−2^ nm^−1^ sr^−1^ to 0.63 mW m^−2^ nm^−1^ sr^−1^ at the far-red band and reduced from 0.62 mW m^−2^ nm^−1^ sr^−1^ to 0.53 mW m^−2^ nm^−1^ sr^−1^ at the red band.

Based on the analysis above, we designed the list of parameter values shown in Table 4. The fitting windows for far-red SIF retrieval were selected to cover the weak water vapor absorption lines and Fraunhofer lines between 712 and 758 nm; 758 nm was taken as the final wavelength, only the starting wavelength (λ_1_) of the fitting window was changed, and the width of the window was then determined. The fitting windows for red SIF retrieval were selected between 682 and 697 nm to cover the oxygen absorption lines. Then, 682 nm was taken as the starting wavelength, and the fitting window was determined by the ending wavelength (λ_2_).

We took the intensity of SIF at a specific wavelength (depending on μ_h_) simulated by SCOPE as the true values, which were compared to the SIF retrievals. The root mean square error (RMSE) was used to evaluate the influence of the parameters, and the optimized parameters were determined for these settings with the smallest RMSE.

## 3. Results

### 3.1. Influence of Empirical Parameters on SIF Retrievals

The range of retrieval RMSE when different values were set for a parameter stands for the necessity of optimizing this parameter. In this study, we selected the median value of each parameter in Table 4 as a default value. Then, for each parameter, the RMSE of different values were recorded when other parameters were fixed as their default value in order to clarify the influence of different parameters on retrieval accuracy (Figure 3).

The fitting window of SIF retrieval (represented by λ_1_ and λ_2_) notably affected the accuracy of SIF retrieval, both in the far-red and red fitting windows. Improper setting of the fitting window greatly increased the RMSE to 1.48 mW m^−^^2^ sr^−^^1^ nm^−^^1^ and 8.79 mW m^−2^ sr^−1^ nm^−1^ for far-red and red SIF retrieval, respectively. However, an RMSE of 0.59 mW m^−2^ sr^−1^ nm^−1^ and 0.87 mW m^−2^ sr^−1^ nm^−1^ could be achieved when the optimal fitting window was selected. The changing range of RMSE can be 104.51% and 295.52% of the average RMSE in different far-red and red fitting windows, respectively. The polynomial order (n_p_) and the feature vectors (n_SV_) also have a great influence on the retrieval. For far-red SIF retrieval, the RMSE changed within 0.57–1.09 mW m^−2^ sr^−1^ nm^−1^ with different n_p_ (66.67% of the average RMSE), and when a different n_SV_ was set, the RMSE varied between 0.57 and 0.96 mW m^−2^ sr^−1^ nm^−1^ (47.56% of the average RMSE). For red SIF retrieval, the influence of n_p_ and n_SV_ was greater. An RMSE within 0.76–3.27 mW m^−2^ sr^−1^ nm^−1^ (138.67% of the average RMSE) and 1.02–3.36 mW m^−2^ sr^−1^ nm^−1^ (121.62% of the average RMSE) can be obtained when selecting different n_p_ and n_SV_, respectively.

On the contrary, the impact of μ_h_ and σ_h_ was negligible, and the changing ranges of RMSE using different μ_h_ and σ_h_ were no more than 5% of the average RMSE value for both far-red and red SIF retrieval.

Thus, the fluorescence function was fixed according to the study of Du et al. [18], i.e. μ_h1_ is 740 nm, σ_h1_ is 21 nm, μ_h2_ is 692 nm and σ_h2_ is 9.5 nm. The impacts of λ_1_, λ_2_, n_p,_ and n_SV_ are evaluated in a more detailed below.

#### 3.1.1. Influences of Fitting Window and SNR on SIF Retrievals

SIF retrieval is based on the filling effect in the absorption lines and the best fitting window may be dependent on the SNR. Thus, we calculated the RMSE of SIF retrieval using different fitting windows and simulation datasets with different SNRs (Figure 4).

The retrieval accuracy was low in narrow fitting windows. For the far-red band, the RMSE was greater than 0.70 mW m^−2^ sr^−1^ nm^−1^ when λ_1_ was greater than 740 nm, and the accuracy did not appear to be significantly improved with the increase in the SNR in narrow fitting windows. In wider fitting windows (λ_1_ < 740 nm, λ_2_ > 691 nm) that contain enough absorption lines, a relatively low RMSE of less than 0.70 mW m^−2^ sr^−1^ nm^−1^ can be achieved, even when the SNR is 322, by appropriately adjusting other parameters in Table 4.

Figure 4 shows the best retrievals of SIF with different fitting windows, but it has not been tested whether the best retrievals in these fitting windows are robust with the changing of n_SV_ and n_p_. To select an optimal fitting window that can achieve an accurate and robust retrieval using SIFIS data, the simulation dataset with an SNR of 322 was used to investigate the distribution of RMSE in different fitting windows (Figure 5).

For far-red SIF retrieval, the RMSE varied within wide limits when λ_1_ was greater than 735 nm, which increased the difficulty of the optimization of empirical parameters in these fitting windows. When λ_1_ was 735 nm, the RMSE was the smallest and most robust with a value of 0.93 ± 0.44 mW m^−2^ sr^−1^ nm^−1^. For red SIF retrieval, the varying range of RMSE continued to decrease as the fitting window widened. A fitting window of 682–697 nm achieved an optimal RMSE (0.95 ± 0.38 mW m^−2^ sr^−1^ nm^−1^).

#### 3.1.2. Influence of Polynomial Order and the Number of Feature Vectors on SIF Retrievals

The influence of the polynomial order is clarified in the optimal fitting window selected in Section 3.1.1 (Figure 6). As can be seen, the RMSE decreased first and then increased with the increase in the polynomial order. The second-order polynomial was a better choice in both far-red and red fitting windows, with an RMSE of 0.82 ± 0.36 mW m^−2^ sr^−1^ nm^−1^ and 0.71 ± 0.16 mW m^−2^ sr^−1^ nm^−1^ for far-red and red SIF retrieval, respectively.

The RMSEs of different numbers of feature vectors were evaluated (Figure 7). The accuracy of SIF retrieval decreased first and then increased with the increase in n_SV_ in both the far-red and red fitting windows as too many feature vectors over-fit the random noise of the sensor. For far-red SIF retrieval, n_SV_ was set as 4 with a minimal RMSE of 0.63 mW m^−2^ sr^−1^ nm^−1^, while for red SIF retrieval, n_SV_ was set as 7 with a minimal RMSE of 0.53 mW m^−2^ sr^−1^ nm^−1^.

Except for the analysis above, we also investigated whether different fitting windows and SNR affected the optimal setting of n_p_ and n_SV_ (Figure 8).

Our results showed that in far-red fitting windows, there was an obvious positive linear correlation between window width and the optimal n_p_ and n_SV_, with an R^2^ of 0.71 and 0.46 respectively, which means that a larger number of feature vectors and polynomial orders are preferable in wider fitting windows. In the red band, the SIF signal was weaker and more affectable by the random noise. Thus, for red SIF retrieval, although the optimal n_p_ and n_SV_ still had a positive linear correlation with the width of fitting windows, the correlation between λ_2_ and the optimal n_p_ or n_SV_ was lower than that of far-red fitting windows with the R^2^ of 0.24 and 0.38 for the relationship between λ_2_ and the optimal n_p_ and the optimal n_SV_, respectively.

In addition to the influence of the fitting windows, the scatter points in a certain fitting window also represent the results corresponding to data with different SNR. In some fitting windows, such as 720–758 nm, there was more than one optimal value of n_SV_ and n_p_ for data with different SNR, which indicates that the SNR also affects the optimization of the empirical parameters. However, in our selected fitting windows (682–697 nm for red SIF retrieval and 735–758 nm for far-red SIF retrieval), there was only one combination of optimal n_p_ and n_SV_ which was not affected by the SNR.

### 3.2. End-to-end SIF Retrievals of the Optimal Empirical Parameters

End-to-end SIF retrieval for TECIS-like simulations was carried out to test the retrieval ability of the selected optimal combinations of empirical parameters. We calculated the mean and standard deviation of SIF retrieved under 1280 different atmospheric conditions (SIF_retrieved_) and took 126 SIF values simulated by the SCOPE model as true values (SIF_true_). Here, we show end-to-end SIF retrievals using the optimal empirical parameters for SIFIS onboard TECIS-1 (Figure 9).

The scatter points distribute very close to the 1:1 line for both simulations in the far-red and red band. For far-red SIF retrieval, the optimized parameters setting in the data-driven algorithm are given with a 735–758 nm fitting window, second-order polynomial, and four feature vectors. When the SNR is 322, the standard error of SIF retrieval is 0.62 mW m^−2^ sr^−1^ nm^−1^ and the RMSE value is 0.63 mW m^−2^ sr^−1^ nm^−1^. For red SIF retrieval, the optimized parameters are second-order polynomial and seven feature vectors using a fitting window of 682–697 nm. The standard error of SIF retrieval was 0.44 mW m^−2^ sr^−1^ nm^−1^, and the RMSE value was 0.53 mW m^−2^ sr^−1^ nm^−1^. Furthermore, the error lines of different SIF intensities have almost the same amount under a certain noise level (e.g., SNR = 322), which indicates that the larger SIF levels have a smaller relative error and the absolute error is mainly affected by noise while independent of SIF intensity.

We also evaluated the fitting accuracy of the reconstructed radiance spectra between 735–758 nm and 682–697 nm with and without the filling of fluorescence under an SNR of 322, as shown in Figure 10. Figure 10 also shows excellent fitting accuracy of the forward model to reconstruct the TOA radiance. The residual value was less than 0.11 mW m^−2^ sr^−1^ nm^−1^ for far-red SIF retrieval and less than 0.04 mW m^−2^ sr^−1^ nm^−1^ for red SIF retrieval. The forward model fitted with SIF fit the radiance better with a lower residual error.

## 4. Discussion

Our results showed the variation of the retrieval accuracies using several empirical parameters in the model of the data-driven algorithm. Firstly, as shown in Figure 4 and Figure 5, more accurate retrievals can be achieved using wider fitting windows. This is largely because the broader spectral windows may contain more information about the absorption line [32], while reducing the influence of instrumental noise [19]. However, a wide window is not always correlated with an accurate retrieval, which can be explained by the fact that it is more difficult to accurately fit the variation of radiance in a wider window [19]. Secondly, the retrieval accuracy increased first and then decreased with the increase of n_p_, while the optimal retrieval was achieved when n_p_ is 2 or 3. This is easy to understand, for the low-frequency part of the vegetation reflectance spectrum has a fixed smooth shape, which can be well fitted by a 2- or 3-order polynomial in a fitting window narrower than 30 nm [19]. Besides, the accuracy of retrieval first increases and then decreases as n_SV_ increases from 2 to 20. This is because the variance that can be explained by feature vectors increased with n_SV_ at first and gradually saturated [12]. Most of the variance of the training set can be carried out by our selected feature vectors (99.89% of the variance can be explained by the first four feature vectors at far-red band, while the first seven feature vectors at red band can explain 99.99% of the variance). The subsequent feature vectors can only explain less than 0.2% of the variance, which mostly comes from the random noise of the instrument.

It has to be admitted that our set of parameters is only designed for SIFIS-like data and may not apply to other instruments with different spectral characteristics. For satellites with higher spectral resolution (<0.1 nm), it is better to use a narrower fitting window, which only covers the solar Fraunhofer lines [11,15,18]. Nevertheless, the reasonable RMSE and the ability of the forward model to fit the radiance spectra indicate that accurate and robust retrievals can be achieved using our optimal parameters at both far-red and red bands. It can also be expected that the optimal combination of empirical parameters in this paper can not only be applied to TECIS-1, but also have the potential of SIF retrieval onboard other satellites to be launched, such as the FLuorescence EXplorer (FLEX) [33] and the Tropospheric Emissions: Monitoring of Pollution (TEMPO) [34] geostationary mission.

## 5. Conclusions

In this paper, simulation data were employed to investigate the influence of different empirical parameters on far-red and red SIF retrieval. The fitting window, the polynomial order, and the number of feature vectors were the three main factors that influenced SIF retrieval in both far-red and red band. The retrieval accuracy was relatively higher for a wider fitting window, and an RMSE of lower than 0.7 mW m^−2^ sr^−1^ nm^−1^ was achieved with fitting windows wider than 735–758 nm and 682 nm–691 nm for far-red band and red band, respectively. Moreover, the retrieval accuracy was dependent on the polynomial order and the number of feature vectors. The RMSE decreased first and then increased with the increase in n_p_ and n_SV_ and can be lower than 0.7 mW m^−2^ sr^−1^ nm^−1^ when n_p_ is 2 or 3; such a result can also be obtained when n_SV_ is 3–5 and 4–14 in the far-red and red band with an n_p_ of 2, respectively. According to the specifications of SIFIS onboard TECIS-1, a fitting window of 735–758 nm, second-order polynomial, and four feature vectors were the optimized parameters for far-red SIF retrieval, with an RMSE of 0.63 mW m^−2^ sr^−1^ nm^−1^. For red SIF retrieval, using second-order polynomial and seven feature vectors in the fitting window of 682–697 nm was the optimal choice and resulted in an RMSE of 0.53 mW m^−2^ sr^−1^ nm^−1^. The optimized parameters of the data-driven algorithm can guide the retrieval of satellite-based SIF and are valuable for generating an accurate SIF product of the TECIS-1 satellite after its launch.

## Figures and Tables

**Figure 1 sensors-21-03482-f001:**
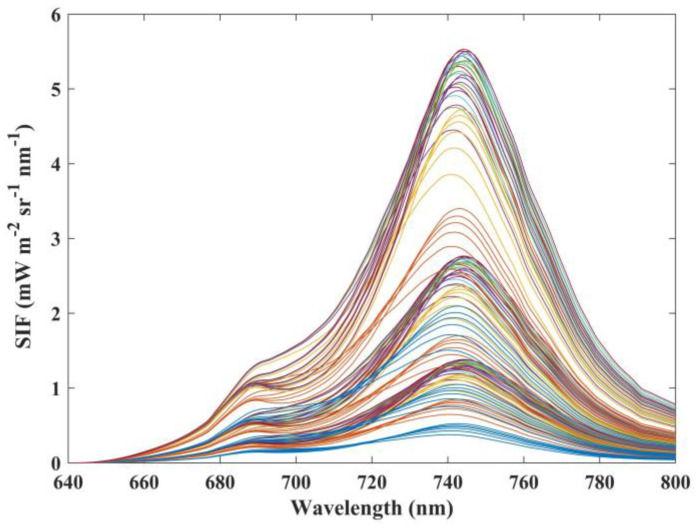
Fluorescence signal at the top-of-canopy (TOC) simulated by SCOPE.

**Figure 2 sensors-21-03482-f002:**
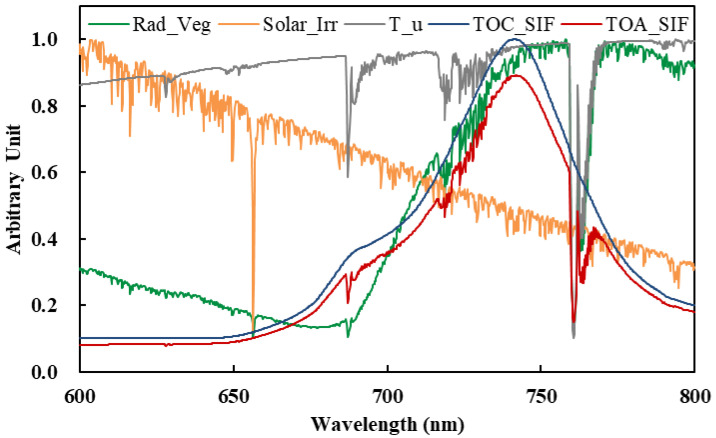
Simulated signals with 0.3 nm spectral resolution and 0.1 nm sampling interval derived from the SCOPE and MODTRAN 5 models, including the top–of–atmosphere (TOA) radiance over the vegetated surface (Rad_Veg), solar irradiance (Solar_Irr), upward transmittance of the atmosphere (T_u), SIF signal at the top of the canopy (TOC_SIF) and atmosphere (TOA_SIF).

**Figure 3 sensors-21-03482-f003:**
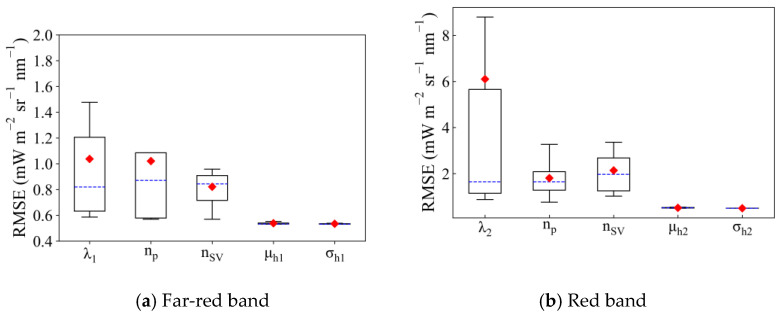
The root mean square error (RMSE) of SIF retrieval when different values for each parameter were set. The influence of each parameter was investigated when other parameters were set as their default values, i.e., μ_h1_ = 740 nm, σ_h1_ = 30 nm, μ_h2_ = 688 nm, σ_h2_ = 10 nm, n_p_ = 3, n_SV_ = 11, λ_1_ = 730 nm, and λ_2_ = 691 nm (SNR = 322). The horizontal bar in blue represents the median, diamonds in red represent the average, and the box bar covers 50% of the RMSE values. RMSE values that do not fall within the upper and lower limits of the boxplot are excluded as outliers.

**Figure 4 sensors-21-03482-f004:**
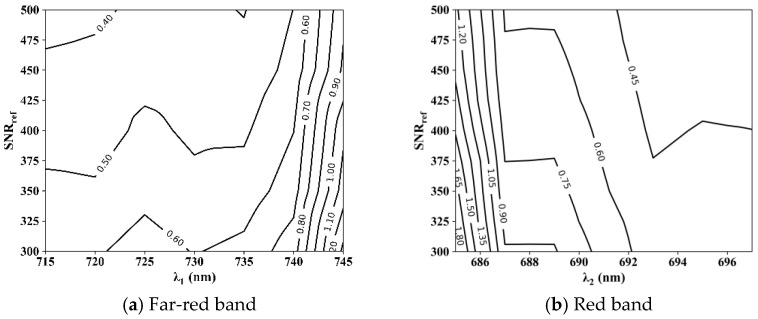
The root mean square error (RMSE) of (**a**) far-red and (**b**) red SIF retrieval using different fitting windows (λ_1_—758 nm at far-red band and 682—λ_2_ nm at red band) and simulation datasets with different SNRs. Other parameters listed in Table 4 are arbitrarily changed, and the minimal value of the RMSE is taken.

**Figure 5 sensors-21-03482-f005:**
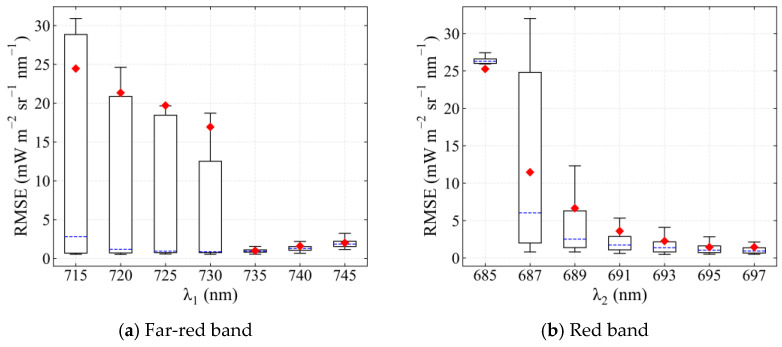
The RMSE of SIF retrieval in different (**a**) far-red and (**b**) red fitting windows (λ_1_—758 nm at far-red band and 682—λ_2_ nm at red band) when the other parameters in Table 4 arbitrarily change (SNR = 322). The horizontal bar in blue represents the median, diamonds in red represent the average, the box bar covers 50% of the RMSE values, RMSE values that do not fall within the upper and lower limits of the boxplot are excluded as outliers.

**Figure 6 sensors-21-03482-f006:**
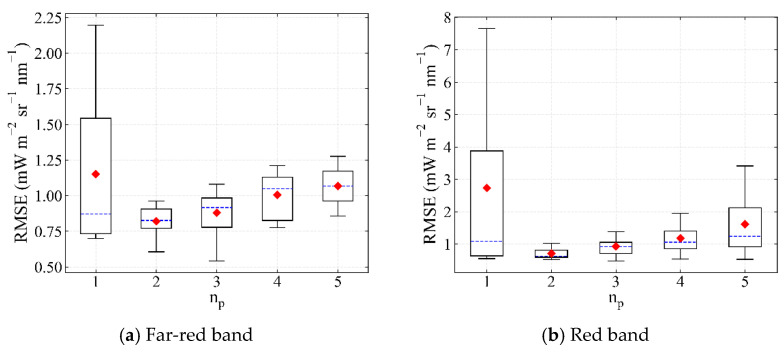
The RMSE of SIF retrieval in the optimal (**a**) far-red (735–758 nm) and (**b**) red fitting windows (682–697 nm) with different polynomial orders when the other parameters in Table 4 arbitrarily change. The horizontal bar in blue represents the median, diamonds in red represent the average. The case where n_p_ is 5 in the far-red fitting window is excluded for the abnormal RMSE value with an average of 17.14 mW m^−2^ sr^−1^ nm^−1^.

**Figure 7 sensors-21-03482-f007:**
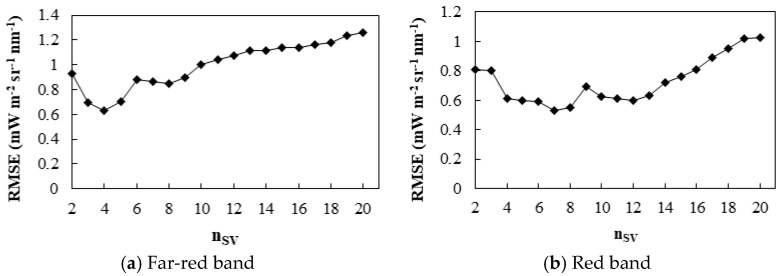
The RMSE of SIF retrieval in (**a**) far-red and (**b**) red fitting windows using different numbers of feature vectors. The order of the polynomial is 2 and the fitting windows are 735–758 nm and 682–697 nm for the far-red and red band, respectively.

**Figure 8 sensors-21-03482-f008:**
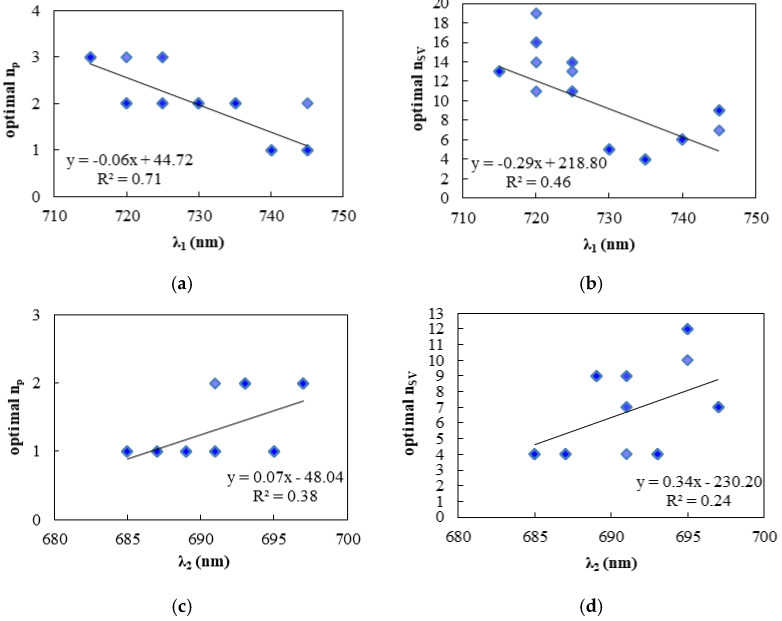
The optimal polynomial order (**a**,**c**) and number of feature vectors (**b**,**d**) of SIF retrieval in different far-red (**a**,**b**) and red (**c**,**d**) fitting windows (λ_1_—758 nm at far-red band and 682—λ_2_ nm at red band). The influence of the SNR is also shown by the number of scattered points in the same fitting window. The optimal parameters were selected with the smallest RMSE.

**Figure 9 sensors-21-03482-f009:**
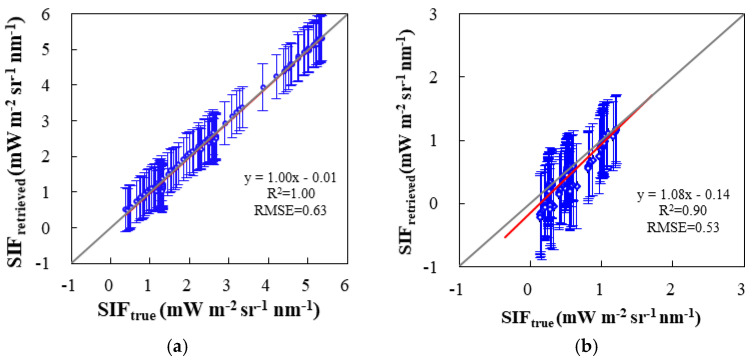
The end-to-end SIF retrieval for the TECIS-1 satellite using (**a**) far-red and (**b**) red fitting windows with optimized parameter setting (SNR = 322). For far-red SIF, the fitting window is 735–758 nm, the order of the polynomial is 2, and the number of feature vectors is 4. For red SIF, the fitting window is 682–697 nm, the order of the polynomial is 2, and the number of feature vectors is 7. The standard deviations of retrieved SIF are depicted by the error bars.

**Figure 10 sensors-21-03482-f010:**
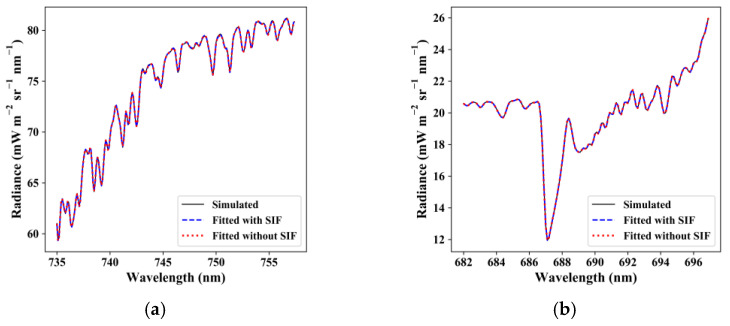
Fitting (**a**,**c**) radiance spectra and (**b**,**d**) residual error using (**a**,**b**) far-red fitting window (735–758 nm) and (**c**,**d**) red fitting window (682–697 nm) when SIF is fitted and not fitted.

**Table 1 sensors-21-03482-t001:** The spectral characteristics of the solar-induced chlorophyll fluorescence Imaging Spectrometer (SIFIS) onboard the First Terrestrial Ecosystem Carbon Inventory Satellite (TECIS-1).

Parameter	Spectral Resolution (nm)	Sampling Interval (nm)	Spectral Range (nm)	Signal-to-Noise Ratio
Value	0.3	0.1	670–780	322 *

* The reference radiance is 10 mW m^−2^ sr^−1^ nm^−1^.

**Table 2 sensors-21-03482-t002:** Look–up table (LUT) for the Moderate-resolution atmospheric TRANsmission (MODTRAN) and the Soil Canopy Observation Photosynthesis and Energy (SCOPE) models.

**Parameters of MODTRAN5**	**Value**
Atmospheric temperature profile	middle latitude summer/winter
Total column water vapor (g cm^−2^)	0.5, 1.5, 2.5, 4
View zenith angle (degree)	0, 16
Final altitude (km)	0.01, 0.05, 1, 2
Aerosol optical thickness at 550 nm (km)	0.05, 0.12, 0.2, 0.3, 0.4
Solar zenith angle (degree)	15, 30, 45, 70
**Parameters of SCOPE**	**Value**
Leaf area index (LAI)	0.5, 1, 2, 3, 4, 5, 7
Fluorescence quantum efficiency (fqe)	0.01, 0.02, 0.04
Chlorophyll content (Cab) (μg cm^−2^)	20, 30, 40, 50, 60, 80

**Table 3 sensors-21-03482-t003:** Comparisons of retrieved SIF (SIF_retrieved_) and true SIF (SIF_true_) simulated using SCOPE model in different experiments. A certain number of feature vectors (n_SV_) and polynomial order (n_p_) were used within the fitting window from the starting wavelength w_1_ to the ending wavelength w_2_. The statistical results included the root mean square error (RMSE), the mean difference (bias) between SIF_retrieved_ and the SIF_true_, the correlation coefficient (r), as well as the slope and intercept of the linear fitting (SIF_retrieved_ = slope * SIF_true_ + intercept). The radiance units (denoted by *) are mW m^−2^ nm^−1^ sr^−1^.

Exp.	w_1_	w_2_	n_p_	n_SV_	RMSE *	bias *	r	slope	Intercept *
1	755	778	2	7	1.18	−0.34	0.77	0.87	−0.03
2	747	780	4	7	0.86	0.01	0.87	0.99	0.03
3	724	747	2	10	0.80	0.33	0.91	1.01	0.29
4	715	748	3	9	0.69	0.14	0.93	0.95	0.25
5	720	758	3	11	0.61	0.24	0.96	0.98	0.29
6	735	758	2	4	0.63	−0.03	0.93	1.00	−0.01
7	735	758	4	4	0.78	0.25	0.91	1.04	0.16
8	735	758	2	15	0.84	−0.01	0.88	1.03	−0.07
9	735	758	2	20	0.92	−0.01	0.87	1.03	−0.09
10	682	697	2	5	0.60	0.19	0.57	1.24	0.06
11	682	697	2	7	0.53	0.11	0.60	1.08	−0.14
12	682	697	2	10	0.62	0.26	0.53	1.10	0.20
13	682	697	5	4	0.53	0.01	0.57	1.13	−0.03
14	682	697	1	15	0.56	−0.18	0.51	0.98	−0.17

**Table 4 sensors-21-03482-t004:** The parameters involved in SIF retrieval using the data-driven algorithm.

Parameter	Description	Range	Step
λ_1_ (nm)	Starting wavelength of far-red fitting window	[715,745]	5
λ_2_ (nm)	Ending wavelength of red fitting window	[685,697]	2
n_p_	Polynomial order	[1,5]	1
n_SV_	Number of feature vectors	[2,20]	1
μ_h1_ (nm)	The central wavelength of h_f_ at the far-red band	[735,745]	1
σ_h1_ (nm)	The standard deviation of h_f_ at the far-red band	[20,40]	1
μ_h2_ (nm)	The central wavelength of h_f_ at the red band	[683,693]	1
σ_h2_ (nm)	The standard deviation of h_f_ at the red band	[9,11]	0.5

## Data Availability

Data is contained within the article.

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
