# Peer review of "Optimizing the Empirical Parameters of the Data-Driven Algorithm for SIF Retrieval for SIFIS Onboard TECIS-1 Satellite"

_sensors, 2021, doi:10.3390/s21103482_

Round 1
Reviewer 1 Report
The main purpose of this manuscript, to investigate the influence of different empirical parameters and optimize them for the data-driven algorithm of SIF retrieval, is important and timely. The logic and structure of the manuscript are reasonable. However, there are several parts of this papers can be further improved, as follows: ##General comments: In the manuscript, the authors optimized the parameters, but did not compare the accuracy results of SIF retrieval before and after optimization. The comparison is needed to quantify the effects of parameter optimization. In the results section, the author elaborated on the influences of different parameters on the accuracy of SIF retrieval, but did not discuss the reasons for these influences. An in-depth mechanical explanation of the causes of these influences will improve the manuscript. ##Specifics comments: 1. Line 171 to 175: Why only change the wavelength on one side of the fitting window? Does the same width of the window but different starting and final wavelengths have an influence? 2. Figure 3 and 5: How do you define the outliers? Unclear. 3. Line 201: “the average RMSE”? You should show it instead of “the median RMSE” in the figures , if they were the different. 4. Line 230: “but not the robust results”? What’s the meaning? 5. Line 271 to 273: “the correlation between λ2 and the optimal np or nSV is reduced; the R2 decreases to...”? “Reduced” and “decreased”? Compare to what? Please rephrase. 6. Line 274 to 277: Why you said that more than one optimal np or nSV indicated SNR’s effects?
Reviewer 2 Report
The manuscript quantitatively investigates the influence of the data-driven algorithm’s empirical parameters, which can help guide the retrieval of satellite-based SIF. The idea is very impressive, I think it just need a minor revision before accepting it. It is suggested the following suggestions to be considered.
- The motivation and innovation of this manuscript should be highlighted in introduction.
- 1. For the retrieval of satellite-based SIF, the review is incomplete. Some traditional retrieval methods should be mentioned including 1) Deep Learning Based Retrieval of Forest Aboveground Biomass from Combined LiDAR and Landsat 8 Data," Remote Sens.-Basel, vol. 11, no. 12, p. 1459, 2019. doi:10.3390/rs11121459.
- For the experimental results, please give some discussions to explain the merits and drawbacks of the proposed method.
Reviewer 3 Report
The Paper is well written, the authors used the correct methodology analysis, the results are presented and discussed adequately. I believe that the paper can be published, as the only comment I would have added in the conclusions some references to other new and future generation sensors that can allow the same type of applications and research. Another discussion note to add would be related to the possibility of analysis with respect to aquatic vegetation and to what extent the different types of substrate underlying the vegetation can affect the signal and the estimation capacity.
